# Influence of Climate Change on Chestnut Trees: A Review

**DOI:** 10.3390/plants10071463

**Published:** 2021-07-16

**Authors:** Teresa R. Freitas, João A. Santos, Ana P. Silva, Hélder Fraga

**Affiliations:** 1Research Centre for the Research and Technology of Agro-Environmental and Biological Sciences (CITAB), University of Trás-os-Montes e Alto Douro (UTAD), P.O. Box 1013, 5000-801 Vila Real, Portugal; jsantos@utad.pt (J.A.S.); hfraga@utad.pt (H.F.); 2Research Centre for Agricultural Sciences and Engineering, Department of Agronomy, University of Trás-os-Montes e Alto Douro (UTAD), P.O. Box 1013, 5000-801 Vila Real, Portugal; asilva@utad.pt

**Keywords:** adaptation measures, *Castanea*, *Castanea sativa*, climate impact, climate change, Europe, Mediterranean

## Abstract

The chestnut tree (*Castanea* spp.) is an important resource worldwide. It is cultivated due to the high value of its fruits and wood. The evolution between *Castanea* biodiversity and humans has resulted in the spread of chestnut genetic diversity. In 2019, the chestnut tree area worldwide was approximately 596 × 10^3^ ha for fruit production (Southern Europe, Southwestern United States of America, and Asia). In Europe 311 × 10^3^ t were produced. Five genetic poles can be identified: three in Greece, the northwest coast of the Iberian Peninsula, and the rest of the Mediterranean. Over the years, there have been some productivity changes, in part associated with climate change. Climate is considered one of the main drivers of biodiversity and ecosystem change. In the future, new challenges associated with climate change are expected, which could threaten this crop. It is essential to identify the impacts of climate change on chestnut trees, improving the current understanding of climate-tree interconnections. To deal with these projected changes adaptation strategies must be planned. This manuscript demonstrates the impacts of climate change on chestnut cultivation, reviewing the most recent studies on the subject. Furthermore, an analysis of possible adaptation strategies against the potentially negative impacts was studied.

## 1. Introduction

Worldwide, the chestnut tree (*Castanea* spp., Fagaceae family) is an ecologically, economically, and culturally important resource [1]. Chestnut trees are reproduced using seedlings or vegetative propagation, namely rooted cuttings [2]. These trees can be found in natural and semi-natural forest stands, or managed stands, including traditional orchards and modern plantations [3,4]. They are cultivated for the importance of their fruit and wood. The fruit is used in the preparations of many recipes due to its high nutritional value and can be eaten in two different ways: consumption of fresh fruits or processed products [5]. In fresh conditions, these fruits are characterized by a restricted shelf-life, owed to their high-water activity and starch content [6]. These fruits reveal different types, flavors, or sweetness depending on the variety. For example, the fruits from *Castanea sativa* Portuguese cultivar ‘Judia’ have around 50–80 fruits kg^−1^ [7,8]. Wood is also a chestnut by-product, it is strong but relatively light, having a high density, widely used in the construction of buildings and furniture [5,9,10]. Chestnut woods are rich in water-soluble extract. These conditions are negatively influenced when woods are exposed to air pollutants, promoting wood discoloration [11]. In addition, over the past few decades, the production of edible mushrooms grown underneath chestnut trees was promoted [12].

The current chestnut global distribution is the consequence of natural colonization together with a long history of human intervention. Currently, the chestnut tree area worldwide is approximately 596 × 10^3^ ha for fruit production, being Southern Europe, Southwestern United States of America, and Asia (Japan, Korean Peninsula, and East China) the main distribution regions [13]. According to Figure 1, in recent decades, chestnut world production has shown an upward trend (56 × 10^3^ t yr^−1^) [14].

Nowadays, seven known species of chestnut trees developing in Subtropical, Mediterranean, and temperate forests in the Northern Hemisphere are known [15]. The Chinese chestnut (*Castanea mollissima*), Japanese chestnut (*Castanea crenata*), American chestnut (*Castanea dentata*), and European chestnut (*Castanea sativa*) are widely cultivated owing to the economic relevance for their fruits [1,2]. In 2019, the world’s chestnut production quantity was approximately 2.5 million t, dominated by the production of *Castanea mollissima*, in Asia (2005 × 10^3^ t), followed by *Castanea sativa* in Europe (311 × 10^3^ t). According to the latest available reports, the world main producers, by quantity, were China (77%), Spain (8%), Turkey (3%), the Republic of Korea (2%), Italy (2%) and Portugal (1%) [13] (Table 1). Regarding the cultivated area, China also leads the rank (330,370 ha), followed by Portugal (38,870 ha) (Table 1). As shown in Table 1, China, Turkey, and Spain are the countries with the highest productivity worldwide (>5 t ha^−1^).

In Europe, chestnut trees are documented since Ancient Greece and the Roman Empire [15]. As above mentioned, *Castanea sativa*, commonly known as European sweet chestnut, is the most important species in Europe, mainly located in Western and Southern Europe (Figure 2) [16,17,18,19].

Throughout the centuries, European production has suffered a sharp decline (Figure 3). In 1960, this decline was due to the forest area production replacement by others crops, such as potatoes and cereals. Furthermore, the rural population significantly declined in many regions and most of the ancient chestnut’ groves suffered from abandonment, natural dieback, pests, and diseases [5]. However, during the last decades, chestnut production in Europe has been increasing (Figure 3).

Currently, the species covers more than 2.5 million ha of land. Portugal is the largest European chestnut producer by area (ha) and Spain is the largest European chestnut producer by quantity (t) (Table 1) [3,14]. Moreover, Spain is the country with the highest productivity (>5 t ha^−1^), while Portugal and France show the lowest productivity values (<1 t ha^−1^) (Table 1). This low productivity is can be explained by aged orchards and many isolated trees.

Sweet chestnut trees are usually found at elevations ranging between sea level and 1800 m, although elevations between 700 and 1000 m give the best conditions for fruit production [9,10]. This deciduous species optimally develops in regions with annual mean temperatures between 8 °C and 15 °C, and annual rainfall ranging from 600–700 mm to 1500−1600 mm [3,12,13,16]. The previous conditions highlight its preference for warm and humid temperate climates. This explains its prevailing location in high-elevation areas in Southern Europe, as Mediterranean-type climates are commonly excessively warm and dry for their optimal growth and development.

Chestnut ecosystems are currently threatened by different stress factors (natural or anthropogenic) such as climate change, abandonment of traditional orchards, wildfire, and an increased incidence of pests and diseases [5]. Climate is considered one of the main drivers of biodiversity and ecosystem change. With climate change, modifications in crop microclimate conditions are projected to occur, with implications in the suitability of a given region to grow a specific crop [3,20]. Climate change may modify the physiological and reproductive cycles of species, like anticipating or delaying phenological timings, with implications on yields and fruit quality characteristics. As an illustration, leaf expansion and seed enlargement phases during histogenesis are strongly forced by thermal accumulation. The patterns of pests and diseases associated with chestnuts may also shift with climate change [1,10,20]. Moreover, the quality parameters and chemical composition of chestnuts are largely influenced by climate conditions [21]. According to [22], tree growth is primarily regulated by temperature and precipitation and, secondarily, by soil moisture, solar radiation, and air humidity. Growth is closely linked to climate parameters, thus hinting at its high sensitivity to climate conditions and vulnerability under changing climates. Some climate change facts suggest warmer temperatures and longer growing seasons in the future, accompanied by more frequent and intense extreme weather conditions, such as severe rainfall events, droughts, or heatwaves [9,23,24]. The aforementioned future conditions cause damages to crop in the upcoming decades [25]. For the chestnut ecosystems in the Iberian Peninsula, climate change may represent a major threat, leading to significant losses of goods and ecosystem services [3].

According to the previous lines, it is vital to recognize the impacts of climate change on chestnut trees, improving our current understanding of climate-tree interconnections. Assessing climate change projections under different anthropogenic radiative forcing scenarios, their potential impacts on chestnut tree growth and development, chestnut yield and quality attributes, as well as the identification of suitable and effective adaptation measures, are of foremost relevance for the future sustainability of chestnut cultivation. The present review aims to provide some clues on how climate change may impact chestnut tree cultivation in Europe, as well as to offer an overview of the possible adaptation measures (short- and long-term) that are currently available for chestnut growers. After this introductory section, in Section 2, the Chestnut Tree and Climate Influences are discussed. Section 3 is devoted to the Climate Change Projections and Chestnut Growing Conditions. Lastly, Section 4—Adaptation Strategies, divided into 2 sub-section: Short-term and Long-term, is analyzed.

## 2. Chestnut Tree and Climate Influence

The chestnut species has a good capacity for dynamic colonization, associated with large adaptability, resulting from their genetic and physiological characteristics [26,27]. Geographical parameters, such as latitude and elevation play an important role in chestnut tree cultivation. This cultivation is roughly within the latitude belt from the 27° N (Canary Islands, Spain) to the 53° N (south of the United Kingdom, UK) parallel [9,10,28,29]. The Azores archipelago (Portugal) corresponds to the western limit (25–31° W) of the production of *Castanea Sativa* [30]. Usually, chestnut cultivation is found at elevations ranging between sea level and 1800 m [9,28]. In Southern Europe, there are different elevations with environmental conditions suitable for chestnut development, from high-altitude terrains (e.g., at 1600 m in the Sierra Nevada, Spain) to low-land coastal areas, where thermal amplitudes are relatively low [16,28,30]. The photosynthetic rate is strongly affected by the geographical location, the maximum photosynthetic rates can be found above 800 m, with optimal temperatures in the interval of 22–29 °C for the activity in adult trees [9,31,32]. At low altitudes, the photosynthesis rate can decrease approximately 40%, owing to abiotic stress [30]. Air temperatures also play an important role in the overall photosynthetic capacity [10].

The vegetative cycle depends on climatic conditions, plant development, and cultural practices. In southern European regions, chestnut tree bud break typically occurs at the April, flowering between June and July, and fruit set and maturity from August to October (Figure 4) [9]. Dormancy, i.e., the inability to initiate growth from meristems and other organs and cells, occurs between December and February [13,33]. Concerning the reproductive cycle, chestnut species is monoecious at the flowering level and the female and male flowers are borne on inflorescences called catkins. Regarding inflorescence, chestnut trees are andromonoecious, because they have two types of catkins, unisexual male catkins and bisexual catkins [15].

Chestnut is a mesophilic species from warm temperate climates since the best conditions for its growth are moderate temperature and humidity [13,31,34] Additionally, chestnut is a moderate thermophilic species well adapted to ecosystems with an annual mean temperature ranging between 8–15 °C, and monthly mean temperatures during its vegetative cycle over 6–8 °C [13,30,31,37]. It tolerates a wide range of climatic conditions, varying from cool and wet conditions in the Atlantic bioclimatic region, to warm and dry conditions in the Mediterranean bioclimatic region [3]. Cooler regions associated with nut crops present values of 60,000 growing degree hours (GDH) [20]. The chestnut tree tolerates well maximum temperatures up to 27–31 °C and endures absolute minimum temperatures of as low as –16 °C and adapts to the environment with monthly mean temperatures over 10 °C, during at least 6 months [13,30,31]. Nevertheless, pollen germination only occurs when temperatures reach 27–30 °C [31]. Being that the degree-days (° D) are the sum of the temperature with a base temperature of 6 °C, it is generally accepted that chestnut regions must have 1900–2400° D between May and October [10].

Temperate-climate trees like chestnuts often require relatively cool winters to fulfill their chilling requirements during wintertime dormancy, which allow proper physiological and phenological development, such as budburst, flowering, fruit set, and maturation. Accumulated exposure to low temperatures enables plants to properly set inflorescence production when warmer temperatures arise in spring [25]. The chestnut tree is associated with moderate chilling accumulation (>90 CP) [20]. Subsequently, the temperature accumulation acts as a booster of tree phenology, regulating release from the endo-dormancy period after the accumulation of adequate cold units during wintertime and the release from the eco-dormancy period, whose duration is dependent on forcing units cumulated from the end of endo-dormancy to flowering stage [25]. For example, budburst timing depends on the exposure to cool temperatures (chilling) to release dormancy, followed by optimal temperatures to promote plant growth in spring [33]. Nonetheless, under the same pedoclimatic conditions, different varieties of chestnut trees may have different phenological timings, which are influenced by genetic characteristics [34]. The levels of individual biochemical compounds are also closely connected with the climatic conditions of fruit growth. According to the studies by [38], performed in Portugal (Trás-os-Montes Region, Portugal), in the cooler areas, fruits showed higher moisture content, total phenols, flavonoids, crude proteins, soluble sugars, and starch content, and a clear prevalence of polyunsaturated fatty acids. The extreme summer or winter temperatures may restrict chestnut tree yield performances. Warmer temperatures anticipate vegetative activity, leading to advancing in phenological stages and promoting the increased incidence of diseases and pests [3]. However, this ancient species may exhibit termoinhibition when the air temperature is above 32 °C, which is frequent during summer [9,31,32]. In the case of drought, the tree appears to be particularly prone to increasing water stress leading to mortality risk [39]. On the contrary, cooler temperatures after bud break may result in late-ripening, thus delaying fruit maturation [10].

Precipitation also represents a key factor for chestnut growth. It is generally accepted that there is wide chestnut plasticity, illustrated by a wide range of precipitation levels between 600 and 1600 mm [40]. On the other hand, the length of the drought period is identified as one of the main climatic limitations for chestnut growth, as it might be severely constrained when more than two consecutive months of drought occur, which is indeed very common in Mediterranean-type climates [16]. The reduction in precipitation promotes soil water deficits and plant water stress, affecting plant growth and development, leading to the production of smaller organs hampering flower production and grain filling, and limiting the size and number of individual leaves [25,41,42]. The reduction in grain filling occurs due to a decrease in the accumulation of sucrose and starch synthesis enzymes, thus potentially influencing fruit quality [41].

Water availability is considered an important resource to improve final yields. [43] suggest that water stress and its effects, such as low flower-setting or fruit-setting, are reflected on plant productivity and fruit characteristics, thus requiring further studies on chestnut irrigation. According to the study by [26], *Castanea sativa* Mill. revealed different patterns of variation in phenotypic expression after cultivation in two different water regimes, with 50% and 90% substrate saturation. They concluded that the restricted water supply reduces plant water potential in terms of both height and weight, the root development increased, whilst leaf area decreased and modified the leaf morphology.

The chestnut trees prosper on plains or very gentle hills and mountainsides, as soils with good drainage better support root structures [9,12]. This perennial species can grow where predominant soils are chromic dystric cambisols, derived from migmatitic and gneissic parent materials [12]. It reveals a poor adaptation to chalky or clay soil but appreciates sedimentary, siliceous, and acidic to neutral soils [9,32,40]. Clay soils strengthen the impermeability and compression of chestnut roots because their roots tend to decay in poorly drained soils [40]. Wind-sheltered locations, with adequate temperatures and precipitation, accompanied by good solar exposure, are the best option for the installation of chestnut crops [9,44]. The depth of the soil must be approximately 40–50 cm, it must also be rich in organic matter (3.3%) and have a pH between 5.5 and 6.3 [12,13,44,45]. The chemical components of total nitrogen (N; 0.1%), phosphorus (P_2_O_5_; 8 ppm), and potassium (K_2_O; 320 ppm) stand out in the soil enrichment [12]. A study by [8], which combines soil and temperature indicators, confirmed that low water soil availability and high temperatures are destabilizing factors for chestnut growth, inducing a loss of plant vigor and making the trees vulnerable to ink disease.

Chestnut tree growth is also affected by solar radiation, affecting both photosynthesis and morphology [46]. Chestnut trees grow in average annual sunlight conditions between 2400 and 2600 h [9]. According to other studies, 75% of the maximal photosynthetic rate is fulfilled at 900–1000 µmol m^−2^ s^−1^, which corresponds to almost half of the full sunlight intensity [30,31]. In addition, other studies showed that the maximal CO_2_ fixation is 10 μmol m^−2^ s^−1^ while it is at around 4.5 mmol m^−2^ s^−1^ for *Castanea sativa* variety at 24 °C transpiration rate [31].

As already been mentioned, the chestnut phenology is influenced by climatic variables that affect plant growth and reproduction. The species ability to adapt to external factors is expressed in the characteristics of each cultivar, phenology, the timing of maturation, fruit ripeness, resistance to stress, resistance to pests and diseases, among others [1,13,25] (Figure 5).

Climatic conditions and geographical factors have triggered significant genetic variations of chestnut trees in Europe. Presently, five genetic poles can be identified: three in Greece, the northwest coast of the Iberian Peninsula, and the rest of the Mediterranean basin [30]. The populations from Greece initiate growth earlier, followed by those from Southern Italy and Southern Spain, while ecotypes from Northern Spain and Northern Italy initiate later [30]. Effectively, these different responses of the plants (phenological, morphological, anatomy, and chemical composition) may be associated with the different local climatic conditions and plant adaptive capacity [7,10].

*Castanea sativa* is considered threatened due to its sensitivity to various diseases, such as ink disease and chestnut blight, which are responsible for the decline of many chestnut orchards in several countries of the world [47]. In Europe, *P. cambivora* Petri and *P. cinnamomi* Rand are responsible for ink disease. Air temperature, air humidity, elevation, and soil can be counted as determining factors for the spread of diseases and pests in the chestnut tree. The temperature variation can act on different biological and evolutionary aspects, in the hosts and parasitoids, can reveal effects on the development and death of pathogenic individuals or reduced productivity [31,48,49]. Very wet springs can be detrimental, promoting the development of ink disease (*P. cinnamomi*) and white root rot disease (*Armillaria mellea*) [44]. Air humidity is the special factor for the establishment, spread, and longevity of (*P. cinnamomic*) [3]. Regarding elevation, a study carried out in the Calabrian region, the development of *Dryocosmus kuriphilus* Yasumatsu occurs earlier at lower altitudes (550 m) when compared to higher-elevation sites (1200 m), but high altitudes have a higher number of individuals than lower altitudes [48]. The accumulation of water on clay soils may promote the proliferation of chestnut ink disease (*P. cinnamomi* Rands or *P. cambivora* Buissman or *Armillaria mellea*), compared to porous soils [40]. *P. cinnamomi* Rand and *P. cambivora* Petri cause the root rot disease, which mainly occurs in wet and poorly drained soils, in particular in silty loam soils [50]. The ink disease (*P. cinnamomi*) and chestnut blight (caused by the fungus *C. parasitica*) are the two most destructive diseases affecting *Castanea sativa* and *Castanea dentata* [51]. Hybrid progenies, segregating for *P. cinnamomi* resistance, have been obtained and extensively studied to understand the chestnut resistance mechanisms to ink disease [52]. The rise of temperature in the Mediterranean Basin favors the spread of *C. parasitica* and reduces the systematically acquired resistance of the host trees [40]. In the early 20th century, the massive mortality of *Castanea sativa* in southwestern Europe, which was caused by different species of *P.* spp., led to the introduction of genes of the Asiatic species *Castanea crenata* and *Castanea mollissima*, where rootstocks that are resistant to this fungus resulted [53].

## 3. Climate Change Projections and Chestnut Growing Conditions

The impacts of climate change, which include increased climate variability, have gradual effects on food production, society, and the economy [23,25]. Agriculture is one of the sectors most sensitive to changes in climate and the corresponding impacts. For agriculture, the precipitation and temperature patterns, as well as their distribution throughout the year, and the incidence of extreme weather events, are the most critical variables in sector sustainability [23,54]. Over the last century, a vast literature has revealed that change in the timing of natural events, such as the rise in spring temperatures, has been affecting different plant species around the world [54]. The increasing temperatures, and strengthened intensity and frequency of extreme meteorological events, have augmented the exposure and vulnerability of each specific crop to climate change risks [24,55]. Since the second half of the 19th century, a rise in global mean air temperature has been recorded (currently almost 1.0 °C), accompanied by more frequent and severe weather extremes, largely driven by anthropogenic radiative forcing [56,57]. Heavy rainfall events have become more frequent, whereas cold temperature extremes have become less frequent during the second half of the 19th century [56,58]. Extreme events are considered as sources of crop yield variability and health vulnerability [59]. Furthermore, an increase in atmospheric CO_2_ concentration was been documented. This increase may stimulate productivity growth, but combined with other factors, such as high temperatures (heat stress), the altered precipitation patterns (water stress), accompanied by an increasing frequency of extreme weather events, may hinder crop productivity [23,25,54,60]. The increased interannual temperature variability has been influenced the spring and winter periods. The events in spring have been occurring earlier in recent decades [54,60].

The recent climate trends are expected to strengthen in future decades [58]. To assess future climate impacts, multiple climate models (ensembles) are commonly used [25,55]. According to the Intergovernmental Panel on Climate Change (IPCC), these models are driven by likely future scenarios also known as Representative Concentration Pathways (RCP) [61]. The projections follow a set of greenhouse gas concentration and emissions pathways designed to support research on impacts and potential policy responses to climate change [20,62]. The climate change scenarios can be useful for analyzing agricultural changes due to the increased greenhouse gas emissions and corresponding climate change [24]. Four emission scenarios are usually used, ranging from the less severe to the most severe scenario (RCP 2.6, RCP 4.5, RCP 6.0, and RCP 8.5). The RCP 8.5 scenario corresponds to the pathway with the highest greenhouse gas emissions, while the RCP 4.5 is a stabilization scenario and adopts that climate policies, in this instance the introduction of a set of global greenhouse gas emissions prices, are invoked to achieve the goal of limiting emissions and radiative forcing (4.5 W m^−2^ in the year 2100) [62,63]. These scenarios predict a global mean surface temperature increase, of 2.6 °C or 4.8 °C, for RCP 4.5 or RCP 8.5, respectively [61,64]. As reported by [24], all scenarios showed a decrease in European agricultural land by 2080. Another study assessed the climate change impacts on the physiological conditions of chestnut trees under RCP 2.6 and RCP 8.5 [3]. For RCP 2.6, important risks of loss of ecosystems, and their associated functions, will first appear in the south of the Iberian Peninsula and, at an advanced, the stage chestnut ecosystem will begin a survival phase in the same region. For the RCP8.5 scenario, the risk will be especially high in the Mediterranean areas of Southern and Central Spain, and at a later stage, conditions are projected to surpass the chestnut survival threshold.

Recent studies indicated that climate change is affecting the agricultural systems very differently in different parts of Europe. In most European areas, heatwaves and droughts are very likely to increase in frequency, intensity, and duration [24,65]. In Northern Europe, however, productivity is generally expected to increase, and the range of crop species will grow [24]. Some crops that currently grow in southern Europe will become more suitable further north or in higher elevation areas in the south [23,24]. In Southern Europe, higher temperature and recurrent droughts are projected to worsen conditions in a region already vulnerable to climate variability and change, lowering harvestable yields, increase yield variability, and reduce in areas for traditional crops [23,24,66].

The Mediterranean-type ecosystem brings several changes in the climate, such as warmer and dryer conditions, excessive water demand, and extreme rainfall variability [32,66,67]. As stated by [68], in the Mediterranean Basin climate change decade by decade, causing a downward trend in precipitation (−15%) and an annual temperature increase (0.5 °C). According to [66,67], during the last 50 years, Italy recorded a rainfall reduction of about 135 mm. It is expected that in the future the water-stressed regions will grow in all Mediterranean countries. Climate projections identified a significant increase in heat accumulation and a lowering in chilling accumulation in Portugal [20].

Given that cultivation of chestnut trees for fruit production is mainly concentrated in Southern Europe, more specifically in the Mediterranean Basin, climate change may represent an important threat [40]. Temperature affects the main physiological processes, such as plant growth and development rates, phenological timing, productivity, and quality [25]. In some cases, warmer temperatures may bring some benefits, such as increased productivity, with accelerating fruit ripening, provoking the early harvests [23,25]. Additionally, air temperature is strongly associated with the radial growth of tree stems, being the maximum growth rate correlated with both temperature and day length maximum [22]. On the other hand, insufficient chill accumulation may low fruit set, with detrimental consequences on yields [23,25]. On other occasions, the upward temperature trends may reduce water availability and decreases chilling conditions. Water stress may result in a wide range of negative impacts, such as a low flower-setting and fruit-setting, low leaf area, limited photosynthesis, flower abortion, and cluster abscission [25]. Further, warmer climate conditions are more favorable to the proliferation of insect pests [24]. Low humidity and high temperature protect plants from fungal diseases. It will be necessary to adapt to new climatic conditions, e.g., through varietal selection, modifications in sowing dates, changes in fertilization, or pesticide applications [24]. Water and heat stress lessen plant growth and less vigor, also increasing susceptibility to biotic factors such as ink and blight diseases [29]. For some insect species, such as *Mesopolobus tibialis*, the population doubled when exposed to simulations of sudden stress events due to the cold temperatures (adaptive capacity) [49]. A variation in parasitism rates can be an immediate response to intraseasonal temperature variations, climate change, and variations in habitat.

## 4. Adaptation Strategies

The rapid occurrence of climate change has caused disturbances in crops, such as a rise in disease mortality, the emergence of new pests, increased greenhouse gas (GHG) emission, which requires rapid actions in farming management [69,70]. To combat these effects of climate change, mitigation and adaptation strategies are primary instruments to implement. These measures are managed distinctly, due to differences in priorities for the measures and segregated planning and implementation policies at international and national levels [70,71].

The adaptation measures promote the adaptative capacity and reduce vulnerability to climate change effects, at the same time benefit from positive opportunities resulting from climate change. If the proper management of negative impacts was well designed and implemented the climate change may provide competitive advantages to early-adopting growers [25,71]. According to [55], many studies have emphasized the potential for adaptation to reduce costs or increase gains associated with climate change. To create adaptation measures the future socio-economic scenarios and socio-economic scenarios must be considered, to provide a framework for adaptation decision-making for practitioners [23]. With these facts in mind, the efficiency of each measure is ruled by the local specificities and regional-to-local climate change signals, so it may be implemented on a large scale, at a European level, or on a small scale, as a cultivation company [25]. A decisive factor is the cost of operations that can determine the socio-economic viability and future stability of a designated crop and framework [20]. Crop responses to hostile circumstances are powerfully tied to the implemented adaptation measures, such as a change in agricultural practices and crop management (changing in varieties, training systems, sowing dates, fertilizer, pesticide use, irrigation schedule, pruning, phytosanitary treatments, and harvesting) [20,24,25]. Therefore, the potential of the different adaptation strategies may prevent more dramatic changes in the suitability of a given region to grow crops. Adaptation strategies are divided into two main parameters, short-term and long-term, these measures depend mainly on the application time.

### 4.1. Short-Term Adaptation Strategies

Short-term adaptation strategies are actions taken by growers and are identified as simple changes in orchard interventions that can be applied within one or two seasons, such as irrigation management, soil management, and cover crops, cultural practices, protection against extreme weather, and protection of pests and diseases [25].

#### 4.1.1. Water Management

Drought stress in plants can promote a decline in crop production, reduced water availability, drop soil capabilities, and decrease nutrients available to the plant. In the Mediterranean region, the water requirements are increasing due to climate change [72]. To contest this decline, it is necessary to implement adaptation measures to reform water policy and to promote adequate training to farmers and viable financial instruments [73]. An example is investing in rainwater harvesting equipment or irrigation systems [23].

Irrigation is an efficient option to promote species hydration and is a suitable climate change adaptation strategy, while the implementation of an irrigation system brings new costs to companies, which growers should take into account [25]. In Portugal, only 447 ha are irrigated in 2018, and on the newest 835 ha planted within the year 2007–2013, 23% are irrigated. In France, it is frequent to irrigate chestnut orchards below 50 years of age [74]. According to [43], irrigation application based on tree water potential is enough to increase the chestnut production per tree. Furthermore, the fertigation system, i.e., a combination of irrigation and fertilization, can enhance the irrigation of plants, enhance plant nutritional status and chestnut quality. Refs. [43,75] confirmed that irrigation is a factor that promotes the production, given that chestnut production was 22–37% higher in irrigated trees compared to non-irrigated trees. Moreover, some studies have documented the effect of irrigation on the chestnut’s size index, fruit weight, or production per tree. As reported by [43], irrigation increases the commercial value of the chestnuts, specifically, increase its size keeping their nutritional value and sensory characteristics (fruit size, firmness, tastiness, and sweetness), did not negatively affect the chemical composition of the chestnut.

Other investigations indicated that the selection of the type of irrigation system considers several factors, such as water availability and its purity, soil permeability and its water storage capacity, topography, product value, labor costs, energy costs, capital, and technology requirements [74]. Normally, the irrigation systems found in the chestnut crop are drip systems and micro-sprinkles systems. According to [43], comparing the 2 systems, the drip system is less productive and less expensive compared to the micro-sprinkles system. According to [72], the drip system is the distribution of water to the crops that makes use of the most advanced principles of irrigation. Instead, Regulated deficit irrigation (RDI) and Partial Rootzone Drying (PRD) can be a new way of optimizing crop water use efficiency. Deficit irrigation may be considered a sustainable management option at the present conditions, by reducing irrigation requirements and increasing crop water use efficiency, this technique may save 30% water with the same yield as full irrigation [25,72]. The RDI strategy uses the knowledge of the crop response to water stress at different phenological phases to identify the periods when fruit trees are less sensitive [25]. The PRD is alternate drip irrigation, with both sides of the root zone irrigated alternatively with half the water. Moreover, different sources of water may be used too to promote water management, such as treated wastewater and saline water.

#### 4.1.2. Soil Management

Over time, the soil has shown some degradation due to cover crop management and climate change, such as soil erosion, reduced soil fertility water demand by crops. As reported by [70], climate changes will increase the water demand by crops (40–250%), contributing to rising unpredictability for the future accessibility of freshwater. To solve soil water restrictions with adaptation measure is essential, so it is suggested to increase the water retention capacity in the soil, with an application of mulching (can be combined with the application of pruning residues and agroindustry wastes) that reduced evaporation from the bares soils and protect soil erosion [25,70]. The adoption of this technique has received increasing interest in vegetable cropping systems, due to their capability to supply plant nutrients, control weeds, and maintain soil quality [76].

Alternatively, the adoption of cover crops is a promising soil and water conservation practice for the crop’s areas. In general, cover crops that can enhance soil organic carbon is of special interest in these systems, as it can help to build resilience for climate change adaptation while contributing to mitigate global warming through the sequestration of atmospheric carbon [77]. The cover crop rises the surface roughness, reduces the shear stress of runoff, reduces raindrop splash, enhances soil water infiltration, uptake of atmospheric carbon, and fixation in soil, thus increasing the soil organic carbon [78]. This practice is a clever methodology for sustainable cropping systems as they protect and feed the soil, promote nutrient availability and balance, reduce weed pressure and provide habitat for beneficial insects [76]. When removing the cover crops and organic residues, the soil organic carbon decreases and, consequently, soil erosion increases [78]. Soil fertility is also an important adaptation measure to protect soil characteristics and crop development [25]. According to [45], studies on chestnut fertilization and its possible results are still very rare. The internal organic inputs and reduction in soil disturbance increased the soil organic carbon (SOC) is also increased [77].

#### 4.1.3. Cultural Practice

Farmers can adopt adaptive strategies (managerial and technical in nature) to deal with climate change risk. In the case of managerial measures, which contribute to reducing possible negative effects of climate change on productivity, these can be changing crop training, pruning dates, and training system. In pruning, the application should be focused primarily on enhancing within-canopy light distribution (photosynthesis oriented), aeration of the foliage, and good development of bearing shoots [25]. As a suggestion, pruning residues can be used like mulching, for improving soil fertility.

#### 4.1.4. Protection against Extreme Weather

The negative effects of extreme weather events and high solar radiation in crops urge short-term adaptation strategies. Careful clone selection has been employed to select the more performant varieties in response to extreme temperatures, allowing to minimize damages. Furthermore, protective substances, such as foliar sprays have been developed. In the case of chestnut species, silicon has been studied to protect the plant from extreme events. Silicon is recognized as a fertilizer, bio-stimulating plant protection under environmental stress, which activates the plant’s latent tolerance mechanisms [79]. This compound confers resistance to plants under biotic stress conditions through the combination of a physical and chemical defense system that improves resilience at morphological, physiological, and biochemical levels [80]. More specifically, silicon plays an important structural and protective role in the plant’s protection, with low energy costs due to the increase in the rigidity and abrasiveness (increase in the resistance of xylem vessels to drought and heat) of the vegetal tissues [37]. According to [79], plants treated with silicon showed better-recovering capacity when resubmitted to the new period of optimal temperature (25 °C) after the warm period (32 °C). Additionally, Boron is a plant compound that is affected by warmer temperatures and rainfall. According to [21], when boron is in deficit should apply to the chestnut tree to improve the adaptive capacity of the plant to climatic variations.

#### 4.1.5. Protection of Pests and Disease

As said before, the risks of the proliferation of new pests and diseases have increased with climate change and it is necessary to reduce the impacts caused on production. As an adaptation measure, the development and introduction of resistant or less-susceptible crop varieties. Pesticide application should be carefully considered, taking into account the potential impacts on water quality. Silicon also was used to protect the chestnut tree against pathological fungi which attack leaves, roots, and stems. Ref. [47] conducted a study in which they analyze the influence of silicon on chestnut plants infected with *C. parasitica*, which showed that silicon fertilization can reduce the disease severity and the mortality rate of chestnut plants. Ref. [80] also realized another study in which they analyze the influence of silicon on chestnut plants infected with *P. cinnamomi* and confirmed that phenolic extracts from the plants treated with different silicon concentrations have antifungal activity when exposed to bacteria.

### 4.2. Long-Term Adaptation Strategies

Long-term adaptation options are actions taken by growers, sector stakeholders, and decision-makers to adapt to climate change over an extended period, of three or more seasons, the application of these measures may require more costs than Short-Term measures. Some examples of long-term adaptation strategies are provided in the next sub-sections.

#### 4.2.1. Genetic Selection

The expected global climate changes are a great challenge for plants but also confers a good opportunity to respond genetically (by natural or artificial selection) to environmental change [30]. Effectively, there is a close inter-relationship between climate and genetic variability, which could help to understand the adaptation processes of species that will help to manage and ensure crop conservation [15,52].

Working with plants from the Portuguese “Judia” cultivar, ref. [81] concluded that the morphological and phenological differences among ecotypes are related to the small genetic differences and phenotypic adaptations to different climatic conditions. The chestnut tree has high adaptability to abiotic stresses, resulting from genetic and physiological bases [26]. Specific climatic variables influenced the frequency distribution and fixation of several alleles, resulting in local adaptation processes of the populations [15]. Therefore, there is an adaptative variation among populations from extreme conditions that define a plants’ geographical distribution according to its genetic feature. The production of genetically modified organisms is a powerful tool to develop new resistance mechanisms of resistance to the heterogeneity of the climate and the presence of pests and diseases. In this procedure, attention should be paid to potential risks and biosafety concerns. The tolerance to diseases and insects may be a major aspect of the varietal and clonal selection, thereby avoiding the excessive use of pesticides and herbicides [25]. In the case of the chestnut tree, understanding the basic genetic structure of ink disease resistance will increase the accuracy of genomic selection for disease resistance [15,52]. Vegetative propagation disease-resistant cultivars and rootstocks are desirable for growing chestnuts as a crop. Conventional propagation and micropropagation are the vegetatively propagated method, being that micropropagation is more efficient in the large-scale production of individual genotypes [2].

Effectively, genetic diversity provides the fundamental basis for evolution by natural selection and its preservation within and among populations of a species is necessary to safeguard its potential to adapt to future environmental changes [69]. It is expected that growers will need to replace susceptible varieties with more climate-resilient ones [25]. The observation of close inter-relationships existing between climate and genetic variability is essential for frameworks sustainable.

#### 4.2.2. Relocation

The discussions of species translocations and reintroductions with climate change are growing. With climate change predicted in Europe, some crops that currently grow in Southern Europe will become more suitable in Central and Northern Europe or higher elevations areas in Southern Europe [23,24]. In Southern Europe, new crops and varieties may be introduced only if improved varieties will be introduced to respond to specific characteristics of the growing seasons [24]. In the case of the olive tree, warmer conditions in Europe will determine a possible northerly shift of olive tree cultivation into regions where nowadays excessively low temperatures are commonly a limiting factor for olive growth [25]. According to [82], *C. dentata* is currently present in areas that are northwards of the historical range and climate change could facilitate this expansion.

#### 4.2.3. New Crops

The identification of vulnerable areas and sectors and assessments of needs and opportunities for changing crops and varieties are valuable responses to climate change trends [25]. As a consequence of climate change, new climate-proof crops and cultivars should be introduced and tested, which should be able to provide food security, with improved stress tolerance, selecting promising varieties of cereals, grain legumes, and new crops may be introduced in the coming years [72,83]. According to [72] based on an assessment of the farming systems, new varieties and species with improved tolerance to drought and salinity were introduced in the crop rotations of rainfed and irrigated farming systems.

In Europe, rainfall deficit and extreme summer heat can lead to a severe reduction in nut productivity, and in this case, the development of new chestnut cultivars that promote genetic diversity is a sustainable option to promote the local original permanence of the species [15]. Furthermore, the new crops to be cultivated in the Mediterranean region are quinoa (*Chenopodium quinoa*) and amaranth [25,72]. Quinoa has a significant potential for increased production as a new cultivated crop in the Mediterranean region and other areas of the world including North America, Europe, and Asia, also the cultures bean, chickpea, lentil, nigella, camelina, chia, and linseed show potential interest in the yield in the Mediterranean [25,72,83]. This shows that innovative crops, well adapted to the future warmer and dryer Mediterranean region, can be viable alternatives to current crops [25].

## 5. Conclusions

Climate change has influenced the development of plant species, especially chestnut trees at phenological, physiological, biodiversity, and genetic levels. As much as the biodiversity of the chestnut tree is a key factor for the survival and maintenance of the species and the plant reveals acclimatization characteristics, this will not be enough to overcome the new challenges. The development of future climate projections based on feasible future socio-economic storylines provides objective information that can be used in developing suitable adaptation management. In the case of chestnuts, short- and long-term adaptation measures will have to be applied, such as irrigation and/or the application of protective compounds, which have been studied as a means to reduce the implications of climate change. The adaptation potential of the different strategies to cope with climate change impacts is still unclear but is expected that they can be highly beneficial for the agricultural sector, minimizing climate change impacts on the environment and human activities. Effectively, to keep a stable chestnut sector, investment on the part of the farmers, in addition to government support, is necessary.

## Figures and Tables

**Figure 1 plants-10-01463-f001:**
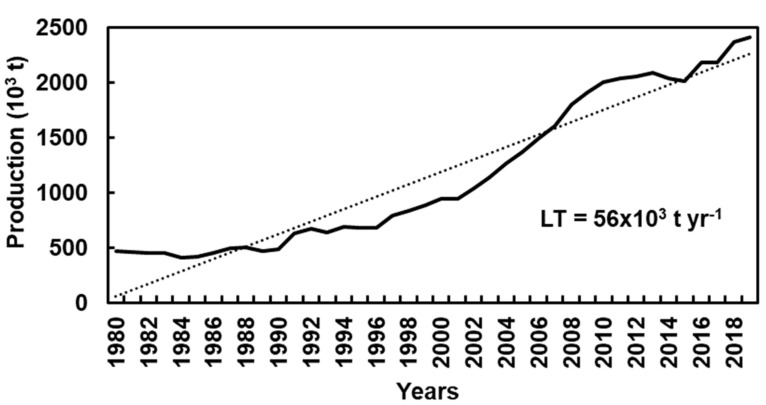
Chestnut production in the world, between 1980 and 2019 [14].

**Figure 2 plants-10-01463-f002:**
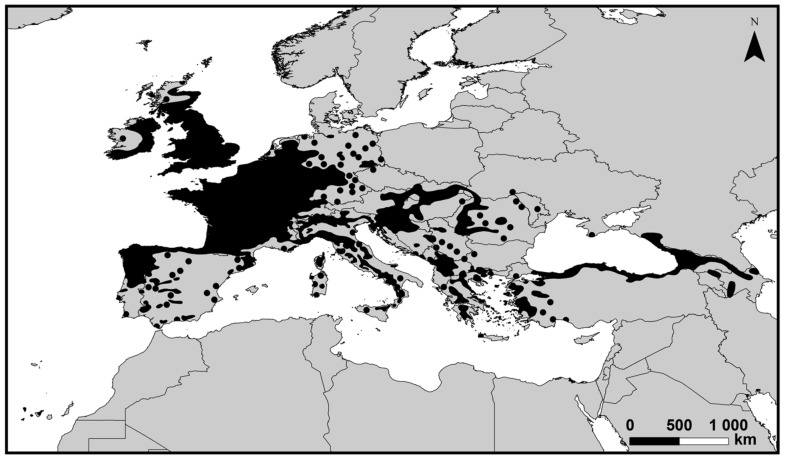
The distribution area of sweet chestnut (*Castanea sativa*) forests throughout Europe (2020), according to [16,17,18].

**Figure 3 plants-10-01463-f003:**
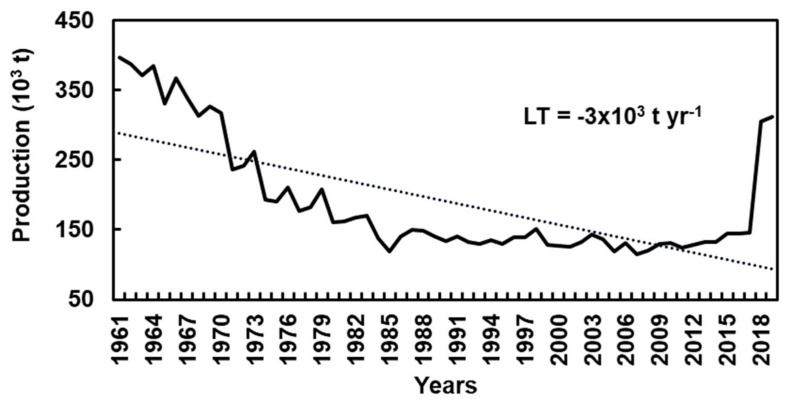
Chestnut production in Europe, between 1961 and 2019 [13].

**Figure 4 plants-10-01463-f004:**
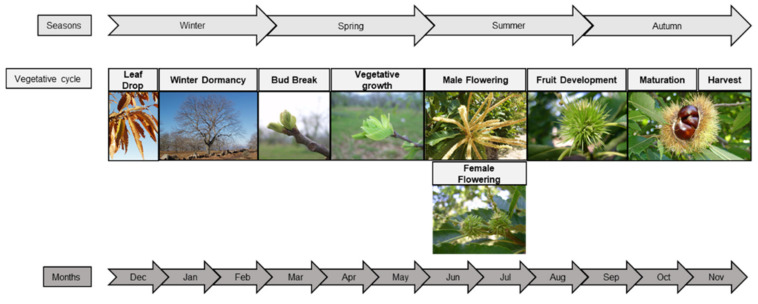
Chestnut tree phenology stages according to the extended BBCH general scale [13,33,34,35,36].

**Figure 5 plants-10-01463-f005:**
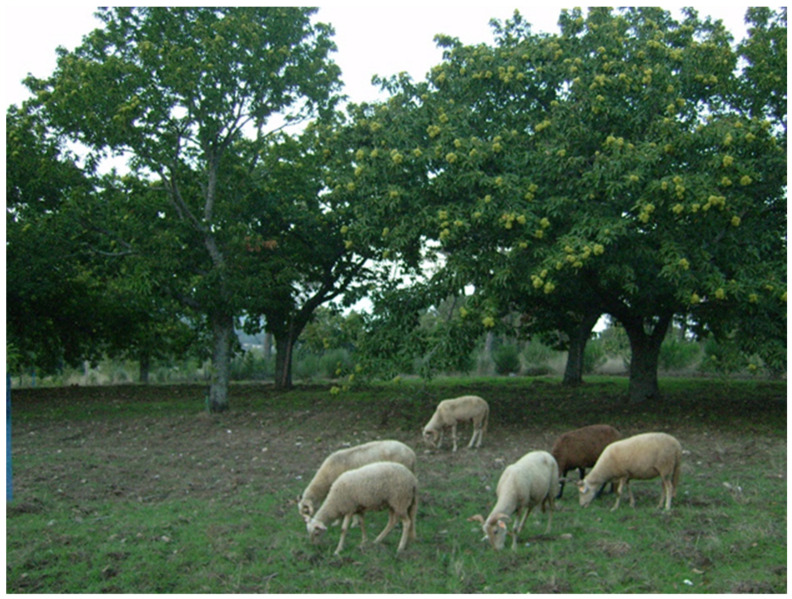
Chestnut orchard associated sustainable agriculture with animal production (photo taken by Bruno Ivo Magalhães).

**Table 1 plants-10-01463-t001:** Production (t), cultivated area (ha), and productivity (t ha^−1^) of chestnut by country, in 2019 [14].

Country	Production (t)	Cultivated Area (ha)	Productivity (t ha^−1^)
Albania	5846	2406	2.4
Bosnia and Herzegovina	2109	1678	1.3
Chile	2848	1263	2.3
China	1,849,137	330,370	5.6
France	7350	8570	0.9
Greece	28,980	8410	3.4
Hungary	200	260	0.8
Italy	39,980	36,280	1.1
Japan	15,700	17,800	0.9
North Korea	12,872	5275	2.4
North Macedonia	1439	1004	1.4
Portugal	35,830	38,870	0.9
Republic of Korea	54,708	32,869	1.7
Romania	40	10	4.0
Slovenia	60	30	2.0
Spain	50,897	37,120	5.1
Turkey	72,655	12,714	5.7
Ukraine	228	82	2.8

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
