# Peer review of "Influence of Climate Change on Chestnut Trees: A Review"

_plants, 2021, doi:10.3390/plants10071463_

Round 1

Reviewer 1 Report

The job is well done. I have no experience in chestnut trees, but I think that the paper could aid farmers of this species to be prepared for climate change future scenarios. 

Only some suggestions:

At line 238 K2O and P2O5 need to be modified as K2O and P2O5

When you explain ink disesase (from line 242 - the first time - to line 283..) sometimes you use Phytophthora cinnamon (line 269; this is a mistake) and other times Phytophthora Cinnamomi that I think correct, as well as Cinnamomi rand....

In my opinion, you should define at the first time exactly specifying that two species of Phytophthora have been found to be responsible for ink disease in Europe [P. cambivora (Petri) and P. cinnamomi Rand]. 

At line 276 Cinnamomi Rabds is a mistake .....Please corretc with Cinnamomi Rand or Rands

When you describe short-term adaptation strategies at point 4.1.4. Protection against Extreme Weather, you write something related to nutrient supply (Silicon and Boron). I think that should be useful to add something about other elements involved in plant protection as potassium or nitrogen, describing their role, In this case, you can change the title of paragraph 4.14. Protection against Extreme Weather (role of some nutrients). 

Reviewer 2 Report

My comments on the manuscript “Influence of climate change on the chestnut trees: A review”, which has been submitted to Plants journal, are presented below.

The manuscript is very interesting. Below I only provide suggestions for the Authors of the work to consider.

In my opinion, the work covers all the issues that the Authors concluded for the purpose of the work. The Authors discussed in detail the issues related to the impact of climate change on the cultivation of chestnuts. In my subjective opinion, there was no photo showing the chestnut plantation. Table 2 contains similar data as table 1? Please correct. It seems that in Table 2 Bolivia does not refer to European countries? I suggest to include non-European countries in one table and European countries in the other, or include one table covering all countries.

I recommend this paper [Manuscript ID plants-1290514] for publication in Plants journal.
